# Exogenous Application of Methyl Jasmonate Increases Emissions of Volatile Organic Compounds in Pyrenean Oak Trees, *Quercus pyrenaica*

**DOI:** 10.3390/biology11010084

**Published:** 2022-01-06

**Authors:** Luisa Amo, Anna Mrazova, Irene Saavedra, Katerina Sam

**Affiliations:** 1Area of Biodiversity and Conservation, Universidad Rey Juan Carlos C/ Tulipán, s/n, E-28933 Móstoles, Spain; 2Biology Centre of the Czech Academy of Sciences, Institute of Entomology, Branisovska 1160/31, 37005 Ceske Budejovice, Czech Republic; anice.manice@gmail.com (A.M.); katerina.sam.cz@gmail.com (K.S.); 3Faculty of Science, University of South Bohemia, Branisovska 1760, 37005 Ceske Budejovice, Czech Republic; 4Departamento de Ecología Evolutiva, Museo Nacional de Ciencias Naturales (MNCN-CSIC), C/ José Gutiérrez Abascal, 2, E-28006 Madrid, Spain; irene.saavedra.garces@gmail.com

**Keywords:** avian olfaction, foraging, herbivore-induced plant volatiles, defense against herbivory

## Abstract

**Simple Summary:**

The tri-trophic interactions between plants, insects, and insect predators and parasitoids are a dominant component of many terrestrial ecosystems. Within these interactions, many predators of herbivorous arthropods use chemical signals provided by the host plants when searching for prey. The exogenous application of methyl jasmonate (MeJA) often induces the release of volatile organic compounds (VOCs) similar to those induced by herbivores in plants. Therefore, it has been used as a method to estimate attraction to VOCs in arthropod and avian predators. In this study, we examined whether potential differences in the composition of VOCs produced by herbivore-induced and MeJA-treated Pyrenean oak trees (*Quercus pyrenaica*) were related to differential avian attraction. Results showed that the overall emission of volatiles produced by MeJA-treated and herbivore-induced trees did not differ and were higher than emissions of Control trees. However, MeJA-treated trees seem to exhibit a higher reaction and release several specific compounds, which may explain the lack of avian attraction to MeJA-treated trees observed in some previous studies.

**Abstract:**

The tri-trophic interactions between plants, insects, and insect predators and parasitoids are often mediated by chemical cues. The attraction to herbivore-induced Plant Volatiles (HIPVs) has been well documented for arthropod predators and parasitoids, and more recently for insectivorous birds. The attraction to plant volatiles induced by the exogenous application of methyl jasmonate (MeJA), a phytohormone typically produced in response to an attack of chewing herbivores, has provided controversial results both in arthropod and avian predators. In this study, we examined whether potential differences in the composition of bouquets of volatiles produced by herbivore-induced and MeJA-treated Pyrenean oak trees (*Quercus pyrenaica*) were related to differential avian attraction, as results from a previous study suggested. Results showed that the overall emission of volatiles produced by MeJA-treated and herbivore-induced trees did not differ, and were higher than emissions of Control trees, although MeJA treatment showed a more significant reaction and released several specific compounds in contrast to herbivore-induced trees. These slight yet significant differences in the volatile composition may explain why avian predators were not so attracted to MeJA-treated trees, as observed in a previous study in this plant-herbivore system. Unfortunately, the lack of avian visits to the experimental trees in the current study did not allow us to confirm this result and points out the need to perform more robust predator studies.

## 1. Introduction

The tri-trophic interactions between plants, insects, and insect predators and parasitoids are a dominant component of many terrestrial ecosystems [1]. As such, they are of particular interest to ecologists and are widely studied [1]. Within the tri-trophic interactions, many predators of herbivorous arthropods use chemical signals provided by the host plants when searching for insect prey [1,2]. Plants naturally contain large amounts of stored constitutive volatile organic compounds (VOCs), and these might be volatilized into the atmosphere by a healthy, unwounded plant depending on their physiochemical properties [3]. However, they are typically volatilized in greater qualities or quantities upon mechanical tissue breakage during herbivore attack [4]. Additional synthesis of novel compounds may be induced by elicitors contained in the saliva of herbivores [5]. After contact with herbivore-specific saliva, the plant synthesizes a hormone methyl jasmonate (MeJA) which mediates the release of so-called herbivore-induced volatile compounds (HIPVs) [6,7]. Induced VOCs may be emitted hours or days after an attack, both from the wounding site only or systemically from undamaged plant leaves [8,9,10,11]. 

Some of these HIPVs can act indirectly as attractants of natural enemies of herbivorous arthropods (e.g., [12,13,14,15]). Insectivorous predators can detect the volatile compounds, track the damaged plant, feed on the arthropods causing the damage, and reduce the abundance of herbivorous insects, thus enhancing the plant’s fitness [16,17,18]. The attraction to HIPVs has been well documented for arthropod predators and parasitoids (predatory mites, parasitoid wasps, predatory bugs, nematodes, etc.—see [1,19,20] for reviews).

Mäntylä and collaborators (2004 [21]) suggested that also other critical predators, insectivorous birds, might be a part of this so-called Cry for Help hypothesis [22]. The authors found that birds discriminate between chemically active and intact trees without seeing the herbivorous larvae or the damage on the leaves [21]. Further, they proposed that the mechanism responsible for the attraction of birds to herbivore-induced trees could be vision [23] or olfaction [24], as herbivore-induced trees differed from uninfested trees both in the reflectance of leaves and the emission of HIPVs [23,25,26]. Subsequent research by Mäntylä and collaborators (2008 [24]) found a positive correlation between avian predation rates of artificial larvae and the quantity of volatiles emitted by mountain birches. Thus, suggesting that olfaction may be the mechanism underlying bird attraction to caterpillar-infested trees [24]. Later, Amo and collaborators (2013 [25]) isolated the chemical and visual cues of trees, and they showed that insectivorous birds safely preferred trees providing a chemical signal over the trees providing only visual signal [25]. Therefore, it seems that olfaction is used by birds to search for food [25,27], although the importance of vision and olfaction in the foraging behavior of birds is not completely understood yet [27,28]. Recent evidence shows that insectivorous birds can detect even small concentrations of HIPVs of herbivore-induced trees that are just developing new leaves [29], or small amounts of volatiles emitted during insect egg deposition [30]. Attraction to caterpillar-infested trees has been studied in different plant-insect-bird systems using ununified methodology [21,23,24,29,30,31,32] and often showing contradictory results. 

The exogenous application of jasmonates, such as MeJA, often induces the release of volatile organic compounds (VOCs) similar to those induced by herbivores in plants [6,11,33,34]. It has been proven that arthropod predators are attracted to MeJA-treated plants, as they are similarly attracted to plants infested by herbivores [2,6,11]. The use of MeJA to simulate insect herbivory has also been used as a method to study avian attraction to HIPVs but with less conclusive results. In the first study under natural conditions, Mäntylä and collaborators (2014 [35]) treated mature mountain birches (*Betula pubescens* Ehrh. Ssp. *Czerepanovii*) with different concentrations of MeJA solutions, and they found that insectivorous birds were not attracted to MeJA-treated trees while they were attracted to caterpillar-infested trees. Later, Saavedra and Amo (2018 [36]) reported that birds were not attracted to MeJA treated Pyrenean oaks (*Quercus pyrenaica* Willd 1805). In contrast, three recent studies showed that wild, insectivorous birds increased their affinity to MeJA-treated mature grey willow shrubs (*Salix cinerea*) [31], English oak *Quercus robur* [11], and *Ficus hahliana* trees [37]. 

Due to the inconsistency of previous results, further research is needed to examine the bird attraction to MeJA-treated trees in different plant-herbivore interactions and to disentangle the causes of the variation in the response of birds to the exogenous application of MeJA. Specifically, it is necessary to determine whether the volatiles released by MeJA-treated trees are similar to those emitted by herbivore-infected trees, as potential differences in the volatile emission between herbivore-infected trees and MeJA-treated trees may explain such differences regarding avian attraction to MeJA-treated trees. 

As mentioned earlier, Saavedra & Amo (2018 [36]) found that 5 mM MeJA-treated Oak trees (*Quercus pyrenaica*) were slightly more frequently visited than untreated trees, but differences were not significant. Unfortunately, whether volatiles emitted by MeJA-treated trees differed from those emitted by herbivore-induced trees was not examined at that time. Here, we present the results of a study aimed to study insectivorous bird attraction to Pyrenean oak trees treated with MeJA and to herbivore-induced trees, and to find the mechanisms underlying avian attraction. We hypothesized that the lack of interest of the birds in MeJA-treated trees in the earlier study [36] could be explained by the lack of similarity of the volatiles emitted by herbivore-induced trees and MeJA-treated trees.

## 2. Material and Methods

### 2.1. Study Area and Species

The experimental study was carried out in May 2021 in a Pyrenean oak (*Quercus pyrenaica*) forest in Madrid province (Sierra de Guadarrama, Central Spain, 40°43′ N, 03°55′ W). In this forest, a population of insectivorous birds breeding in 100 wooden nest-boxes was established in 2017. Nest boxes were occupied mainly by breeding pairs of blue tits (*Cyanistes caeruleus*), and fewer pairs of great tits (*Parus major*). Other insectivorous bird species were observed in the study area at lower densities, including the common blackbird (*Turdus merula*), coal tit (*Periparus ater*), and Eurasian nuthatch (*Sitta europaea*). Tits feed mainly on caterpillars, such as the *Operopthera brumata* or *Tortrix viridiana*, during the breeding period in this region [38,39]. Previous monitoring showed that *T. viridiana* was more abundant on Pyrenean oaks than *O. brumata* (I. Saavedra, personal observation). Therefore, we selected *T. viridiana* as the prey and model species. One week before the experimental study, we hand-collected wild *T. viridiana* caterpillars from oak leaves in the forest and kept them in captivity until they were 5th instar stage. Caterpillars were located in 7 × 6 cm polypropylene containers and fed with fresh leaves of *Quercus pyrenaica*. Water was provided via daily spraying of the leaves.

### 2.2. Experimental Design

We selected 45 adult Pyrenean oak trees with trunks of at least 20 cm in diameter at breast height that were separated by at least 20 m. The experimental trees were alternatively assigned to one of the treatments: MeJA-treated trees (*n* = 15), herbivore-induced trees (*n* = 15), and control trees (*n* = 15). At each tree, we selected one focal branch to which we applied the following treatments. The branches were approximately 0.5–1 m long and 1.5 m above the ground and had no evident signs of herbivory. 

The MeJA treatment consists of applying a 5 mM solution made with distilled water, ethanol, MeJA, and Tween–20. We chose a 5 mM dose as the results of the previous study showed that more insectivorous birds visited the oak trees treated with this MeJA solution than the control and 15 mM MeJA solution treated trees, although differences were not significant [36]. The MeJA-treated trees were prepared by spraying 10 ml of the MeJA solution on a bouquet of leaves (7–10 leaves) on the focal branch of each tree. Each focal branch had around 5–7 bouquets of 7–10 leaves per bouquet, meaning the treatments were applied to 14–20% of the leaves of the focal branch. The treatment was applied every two days for 14 days (i.e., 7 times).

Herbivore-induced trees were prepared by placing 10 individual *T. viridiana* caterpillars on a bouquet of leaves on the focal branch and placed into a green organza sachet (20 × 20 cm) covering the bouquet of leaves (Figure 1). Caterpillars were kept inside the sachet for an entire 14 days. At the end of the experiment, the organza sachet was removed and the caterpillars were released to move to other branches.

Similar empty organza sachets were installed at the rear bouquets of leaves of the focal branches of MeJA and Control treatments to control the effect of the sachet itself (Figure 1). Additionally, control and herbivore-induced trees were sprayed with 10 mL of distilled water according to the schedule of MeJA treatment to avoid differences in the appearance of the moist leaves.

To study the attraction of the insectivorous birds to the trees, we measured the predation rate of birds on artificial larvae. We placed 5 artificial larvae on branches of each tree. The artificial larvae were made of light green plasticine (similar to the natural color of real *T. viridiana* caterpillars, Lepidoptera, Tortricidae, at least according to human-visual perception). The plasticine larvae were similar in size to large fifth instar *T. viridiana* caterpillars (length 15–20 mm, diameter 3–4 mm). The plasticine larvae were attached with cyanoacrylate adhesive glue to the branches of each tree.

We measured the attraction of birds to each tree by checking the number of plasticine larvae with marks indicating predation by birds. This method has been used in previous studies of avian predation [24,31,35,36,40,41,42,43,44,45,46]. The artificial larvae are considered damaged when they had triangle-shaped marks and deep cuts made by bird beaks and when a part of their body was taken by the birds (see [24,35]). Each larva showing a predation mark was replaced with a new one at the same location when larvae were checked during check visits.

The experimental study was conducted under a license issued by the Dirección General de Biodiversidad y Recursos Naturales, Consejería de Medio Ambiente, Ordenación del Territorio y Sostenibilidad, Comunidad de Madrid (Ref. 10/024906.9/20).

### 2.3. Collection of Plant Volatiles

We collected plant volatiles of all experimental trees (*n* = 45), and four additional blank samples, 62 h (mean ± SE = 62 ± 1.05 h) after the beginning of the experiment, i.e., after placing caterpillars on the sachets or applying MeJA treatment. We collected volatiles 62 h after adding the treatments because previous results in another *Quercus* species, the downy oak tree (*Quercus pubescens* Willd), suggest that HIPVs significantly increase from 48 h after infestation by winter moth (*Operophtera brumata*) and green oak tortrix (*Tortrix viridana*) when compared to uninfected control trees (Graham et al. unpublished data, [29]). For each volatile measurement, we removed the organza bag from each branch and removed the caterpillars from the Herbivore-treated trees. We placed the bouquet of leaves where treatments were applied (i.e., MeJA was sprayed and caterpillars present) into a polyethylene terephthalate (PET) bag (Carrefour^®^) closed with a tap. We kept the bag for 60 min, passively filling it with volatiles. Then, we cut one corner of the bag, put in a glass tube for thermal desorption (TD) containing approximately 100 mg of Tenax TA adsorbent (Supelco, mesh 60/80; Bellefonte, PA, USA). The TD tube was connected to a vacuum pump (PAS-500, Spectrex, Redwood City, CA, USA) through a silicon tube. The air was pulled through the TD tube at a flow rate of 200 mL/min. An active volatile sampling was conducted for 15 min, and a total of 3 l of air was pulled through each tube. After that, the TD tubes were removed, closed with silicon caps, and kept refrigerated at 4 °C for about 1 week before analysis by gas chromatography-mass spectrometry. We also measured four blank VOC profiles from empty bags (i.e., background emissions). We sampled the volatiles on two sunny days, between 10:00 and 18:00, with a similar mean temperature (mean ± SE = 25.22 ± 0.59) and humidity (mean ± SE = 46.16 ± 1.04) during sampling. Therefore, there were no significant differences between treatments in the temperature (ANOVA, F_2,42_ = 0.03, *p* = 0.97) or relative humidity (ANOVA, F_2,42_ = 0.01, *p* = 0.99) during sampling volatiles, nor in the time the bag was covering the branch (ANOVA, F_2,42_ = 0.00, *p* = 1.00).

### 2.4. Analysis of Plant Volatiles

Before the experiment, TD tubes were conditioned with a gas chromatograph (5890 Agilent, modified for spike and cleaning TD tubes) for 30 min at 320 °C and a Helium flow rate of 20 mL/min at the Institute of Environmental Technology in Ostrava-Poruba, Czech Republic. The induced volatile samples were analyzed with a gas chromatograph–mass spectrometer (Agilent, GC 7890 + MSD) at the Institute of Environmental Technology in Ostrava-Poruba, Czech Republic. Trapped compounds were desorbed with two-stage thermal desorption using a thermal desorption unit (Perkin-Elmer Turboamatrix TD 300) under described temperatures: Valve = 200 °C, Primary desorption = 250 °C 5 min^−1^, Trap = −10 °C, Secondary desorption = 300 °C min^−1^, Transferline = 200 °C; and flows: Desorption = 40 mL min^−1^, Intel Split = 0 mL min^−1^, Col. = 1 mL min^−1^, Out Split = 10 mL min^−1^; Total split = 9.1%).

Desorbed analytes were injected onto an HP-5 capillary column (30 m, 0.25 mm, 0.25 μm film thickness, Hewlett-Packard) with helium (5N) as a carrier gas with a flow rate of 1 ml min^−1^. The oven temperature program was held at 60 °C for 2 min, then raised to 120 °C at a rate of 10 °C min^−1^, and finally on to 250 °C at a rate of 30 °C min^−1^ with a 5 min delay. The compounds (mono-, homo- and sesquiterpenes, and green leaf volatiles (GLVs)) were identified by comparing their mass spectra with those in the pure standards. Pure chemicals were supplied by Sigma-Aldrich and Supleco, prepared by weighing into methanol:(1)Cannabis terpene Mix B (CRM40937 Supleco, 2000 μg/mL of each component): Limonene (cyclohexene, 1-methyl-4-(1-methylethenyl)-), C_10_H_16_, CAS 138-86-3; β-pinene (bicyclo[3.1.1]heptane-6,6- trimethyl, 2-methylene), C_10_H_16_, CAS 127-91-3; β-Caryophyllene (trans-(1R,9S)-8-Methylene-4,11,11-trimethylbicyclo[7.2.0]undec-4-ene), C_15_H_24_, CAS 87-44-5; Phytol (3,7,11,15-Tetramethyl-2-hexadecen-1-ol), C_20_H_39_O, CAS 7541-49-3; Geraniol (trans-3,7-Dimethyl-2,6-octadien-1-ol), C_10_H_18_O, CAS 106-24-1; (1S)-(-)-Camphor ((1S)-1,7,7-Trimethylbicyclo[2.2.1]heptan-2-one), C_10_H_16_O, CAS 464-48-2; Terpinolene (p-Menth-1,4(8)-diene), C_10_H_16_, CAS 586-62-9; β-Eudesmol ((2R,4aR,8aS)-Decahydro-8-methylene-α,α,4a-trimethyl-2-naphthylmethanol), CAS 473-15-4; (+)-Borneol (endo-(1R)-1,7,7-Trimethylbicyclo[2.2.1]heptan-2-ol), C_10_H_18_O, CAS 464-43-7; cis-Nerolidol (3,7,11-Trimethyl-1,6,10-dodecatrien-3-ol), C_15_H_26_O, CAS 7212-44-4; α-Terpineol (2-(4-Methylcyclohex-3-en-1-yl)propan-2-ol), C_10_H_18_O, CAS 98-55-5; (1S)-(+)-3-Carene ((1S)-3,7,7-Trimethylbicyclo[4.1.0]hept-3-ene), C_10_H_16_, CAS 498-15-7; Linalool ((±)-3,7-Dimethyl-3-hydroxy-1,6-octadiene), C_10_H_18_O, CAS 76-70-6; *p*-Cymene (1-Isopropyl-4-methylbenzene), C_10_H_14_, CAS 99-87-6.(2)Cannabis terpene Mix A (CRM40755 Supleco, 2000 μg/mL of each component): α-Pinene (2,6,6-Trimethylbicyclo[3.1.1]hept-2-ene), C_10_H_16_, CAS 80-56-8; Camphene (3-methylidenebicyclo[2.2.1]heptane), C_10_H_16_, CAS 79-92-5; β-Myrcene (7-Methyl-3-methylideneocta-1,6-diene), C_10_H_16_, CAS 12-35-3; 3-Carene (3,7,7-Trimethylbicyclo[4.1.0]hept-3-ene), C_10_H_16_, CAS 13466-78-9; D-Limonene (1-Methyl-4-(prop-1-en-2-yl)cyclohex-1-ene), C_10_H_16_, CAS 5989-27-5.(3)Single chemicals (Sigma-Aldrich): Caryophyllene oxide, CAS 1139-30-6; Ocimene (3,7-Dimethyl-1,3,6-octatrien), C_10_H_16_, CAS 13877-91-3; cis-3-hexenyl Acetate, C_8_H_14_O_2_, CAS 3681-71-8; Methyl Jasmonate (Methyl 3-oxo-2-(2-pentenyl) cyclopentaneacetate), C_13_H_20_O_3_, CAS 39924-52-2. Emissions were presented qualitatively.

### 2.5. Statistical Analyses

We used STATISTICA 8.0 to perform one-way ANOVA to analyze whether there were significant differences between treatments in the time the bag was covering the branch before measuring volatiles, as well as in the temperature and relative humidity (RH).

We used program R (version 4.0.0; [47]) and *MASS* package [48] to analyze the effect of treatment (MeJA-treated, herbivore-induced, Control) on the total amount of emitted VOCs fitting a Linear Model (package *stats*; [47]). The response variable (total amount of emitted volatiles) was computed by natural logarithm (‘log’ function). For analyses of the effect of treatment (MeJA-treated, herbivore-induced, Control) on individual compounds mean change emission, a Generalized Linear Mixed Models with Template Model Builder (package *glmmTMB*; [49]) was fitted. The Estimated marginal means and multiple contrasts among factors were made using the package *emmeans* [50].

## 3. Results

We did not find any plasticine caterpillar with avian predation marks in any of the three treatments, despite exposing a total of 225 of them for 14 days. This prevented any further analyses of differences in predation between treatments and its link to the chemistry of branches.

Analyses of volatile compounds detected more than 100 different chemicals of various types. We focused on 17 terpenoids differing considerably between the treatments in detail. The amounts of terpenoids released by herbivore-induced and MeJA-treated trees were significantly higher than terpenoid emissions of control trees (ANOVA Chisq = 21.374, Df = 2, *p* < 0.001 [Figure 2]). Total emissions of HIPVs released by MeJA-treated and herbivore-induced oaks did not differ significantly (*p* = 0.53).

Specifically, the application of MeJA resulted in significantly higher production of Trans-β-Ocimene (*p* < 0.05) and Bourbonene (*p* < 0.001) compared to the chemical emissions of control trees (Figure 3). A marginal significance of increased production of β-Ocimene (*p* = 0.054) was also detected. For herbivore-induced trees, the amount of Bourbonene (*p* < 0.001) was significantly different from those emitted from control trees (Figure 3). Within the surveyed compounds, we found only one qualitative difference. MeJA-treated trees did not produce Isolongifolen, which was produced both by herbivore-induced and control trees.

## 4. Discussion

Our results suggest that the overall emission of VOCs produced by MeJA-treated and herbivore-induced trees did not differ, and both were rather different from the emissions of control branches. However, MeJA treatment showed a stronger reaction of experimental branches, as they released several specific compounds in contrast to herbivore-induced trees. Thus, the application of MeJA in manipulative experiments might have caused the overreaction of studied plants and influenced the potential reactions of predators. An alternative explanation is that we did not correctly select the dose of MeJA to match the HIPVs induced by the selected number of caterpillars we used on infest trees.

In a previous study examining avian attraction to MeJA-treated oak trees, results showed that more birds visited the oak trees treated with 5 mM MeJA (7/11) than untreated control trees (4/11) [36]. However, differences were not significant in this previous study, perhaps because this study had low power to detect differences in bird attraction between treatments [36]. To disentangle the questions pointed out in the previous study, we decided to replicate the study with an increased sample size and with adding a new treatment, i.e., herbivore-induced trees. However, in the current study we had an unexpected result, as we did not find any caterpillars being attacked by avian predators. The use of similar artificial larvae for estimating bird attraction to trees has been used successfully in several studies [24,31,36,40,41,42,43,44,45], even within the same bird population [46]. Therefore, the lack of attraction even to Herbivore-infected trees is unlikely due to the use of artificial caterpillars. Another reason might be that the presence of the green organza sachet on the branches scared birds approaching the experimental branches. However, previous studies also used similar sachets or bags to keep caterpillars on a branch in field experiments [24,35] and found birds attacking artificial caterpillars even close to the bags. Therefore, the use of sachets does not seem to be a methodological artifact that may have masked bird attraction to, at least, herbivore-infected trees. A third possible explanation could be that the bird species present in the study area may not discriminate between trees emitting herbivore-induced volatiles or uninfested trees. However, blue tits and great tits are the most abundant insectivorous bird species in the study area, so this explanation can be discarded as both species are already known to be attracted to herbivore-infected trees ([23,24,25,30,51], but see [52]).

Another possible factor impacting our results could be we only treated or infested a small bouquet of leaves on each large experimental tree. It is known that the induced VOCs may be emitted from both the specific place of herbivorous wounding or MeJA application, or systemically from undamaged leaves [8,10,11,53,54,55,56,57,58]. As we measured volatiles from the leaves that were directly herbivore-induced or MeJA-treated, these exact leaves were not accessible to birds because they were inside the sachets. To our knowledge, there are no studies that have examined how localized the response to MeJA exogenous application in *Quercus pyrenaica* is, but the results of a very recent study with a close species, *Quercus robur*, showed that the control branches close to the MeJA-treated branches did not produce more VOCs; i.e., this plant species exhibits a highly localized response [11]. Assuming that the response to MeJA is similarly localized in *Quercus pyrenaica*, it may be that the birds were not able to detect it within the bags. Finally, another possible and nonexclusive explanation of the lack of attraction to herbivore-infected trees is the high availability of food in the forest. We performed the experiment during the spring of an especially rainy year when there was an unusually high abundance of insect prey (I. Saavedra and L. Amo, personal observation).

Despite the fact we could not obtain bird behavioral data in this study, our results from chemical analyses suggest that the slight differences in the emission of volatiles may explain the lack of significant differences in the attraction of insectivorous birds to MeJA-treated and control Pyrenean oaks trees found in a previous study in this plant-herbivore-bird system [36]. Combined results of our current and earlier study [36] are in line with results of a study where mature mountain wild birches (*Betula pubescens* ssp. *czerepanovii*) were treated with 15 mM and 30 mM MeJA solutions and infested with larvae of the autumnal moth (*Epirrita autumnata*) in field conditions. The authors found a significant difference in chemical response to treatment but did not detect a significant difference in attractivity of the MeJA-treated trees in comparison to the control. In both studies, (current and [35]), increased production of α-pinene, both on herbivore-induced and MeJA-sprayed trees, and increased production of limonene and myrcene from MeJA-treated trees was detected.

In contrast, MeJA-treated plants attracted insectivorous birds and predatory arthropods in three studies performed in different systems, using grey willows *Salix cinerea* in the Czech Republic [31], MeJA-treated *Ficus hahliana* in Papua New Guinea [32], and *Quercus robur*, *Carpinus betulus*, and *Tilia cordata* in Germany [11]. Although volatiles were not determined in *Ficus hahliana* trees, Mrazova & Sam [37] measured the HIPVs of grey willows treated with 30 mM MeJA solution and untreated shrubs. The production of α-pinene, β-pinene, 3-carene, limonene, and β-ocimene was higher in MeJA-treated shrubs than in untreated shrubs [31]. In *Quercus robur* and *Carpinus betulus* trans-ß-ocimene increased significantly after MeJA treatment in contrast to the control treatment, but there was no significant effect of MeJA on any compound found in *Tilia cordata*. Unfortunately, herbivore-induced trees were not included in any of the three above-mentioned studies, so it is impossible to say whether herbivore-damaged trees and MeJA-treated trees would differ in their volatile emission and how this could affect avian attraction.

Many studies focusing on arthropod predators have also found differences between the volatiles emitted by plants in response to exposure to jasmonate and those emitted by herbivore-infested plants [34,59,60,61]. For example, MeJA-treated lima bean plants released similar but not identical HIPVs as those released by herbivore-infected plants [62,63]. Although MeJA-treated lima bean plants were still attractive to predatory mites, spider-mite-infested plants were preferred by predatory mites [59]. In contrast, other studies found an attraction of arthropod predators or parasitoids to jasmonate-treated plants (e.g., [7,11,64,65,66]. Differences in the volatile emission between MeJA-treated and herbivore-induced trees can be expected because plant defense responses display a great deal of specificity [67] despite jasmonate representing an important mediator of chemical defense in plants, especially in response to lepidopteran caterpillar herbivory [68]. As a mediator, MeJA is expected to elicit only a more generalized response than the damage caused by any specific herbivore [69]. Our results seem to support it being in line with this expectation as MeJA-treated trees emitted more Trans-β-Ocimene and β-Ocimene than control trees, but no more than herbivore-induced trees. Furthermore, we found that MeJA-treated trees did not produce Isolongifolen in contrast with Control and herbivore-induced trees. Previous results also found that different plant species respond to MeJA treatment by emitting several VOCs, some of which were detected in the herbivore-wounded plants [11], but others were unique to the MeJA treatment [34]. Further experimental studies are needed to disentangle which compounds, and their relative proportions, of these volatile blends can have an important attractive or deterrent role in the discrimination herbivore-infected and uninfected trees.

The variability of HIPVs emissions can be further determined by other factors [19,70]. For example, the emission of HIPVs is known to differ according to the plant species [71,72], the developmental stage of the plant [73], and even the parts attacked by the herbivores [74]. For example, in a previous study with northern red oak (*Quercus rubra*) seedlings, the authors found that the activity of peroxidase isozymes are involved in the tree defense response differed between seedlings treated with MeJA and those infected with caterpillars [75]. The emission of HIPVs also depends on the arthropod species [22,72,76], on the herbivore density [77,78,79], and even on the time course after infestation [80]. Moreover, environmental factors are also known to influence the emission of HIPVs. For example, differences in HIPVs have been found between laboratory and field conditions [6,81]. All of these factors can induce quantitative or quality changes in the volatile blend [82,83,84]. Furthermore, differences in the dosage of jasmonate may also influence the release of volatiles, as well as predator or parasitoid attraction [85]. We decided to use a dose of 5 mM MeJA, as previous results suggested that birds visited oaks trees treated with this dose more often than oaks treated with a higher dose [36]. Further, necrosis on the leaves of the trees treated with 15 mM MeJA solution was observed. However, we have to admit that we do not know how exactly this dose mimics the situation when oak trees are infected by caterpillars. Furthermore, based on previous results with *Quercus pubescens*, where authors found that the emission of VOCs significantly increased from 42 h after infestation (Graham et al. unpublished data, in [29]), we decided to measure the volatiles of our trees after 62 h from infestation or MeJA application. Therefore, differences in the dosage as well as in the timing of the measurement may have provided different results in the analysis of volatiles.

Insectivorous birds are generalist predators that feed on different prey species hosted by different plant species. For them, the signal of the presence of insects should be sufficiently generalistic, yet significantly different from the emissions of VOCs released after simple mechanical wounding. As the jasmonic pathway is involved in both processes, the ability to learn to associate a positive foraging experience with the particular blends of HIPVs seems to be favored over an innate recognition of so many different blends of volatiles ([15,51,86], but see [87]). This ability to associate HIPVs with the presence of food, as well as an innate lack of attraction to infested trees, has been previously demonstrated in two experimental studies with naïve great tits [51,88]. Great tits naïve to foraging in trees were not attracted to herbivore-induced trees, whereas when they experienced foraging experiences, they were able to discriminate between the volatiles of herbivore-infected and uninfected trees, both from native [51,88] and foreign trees species [51]. Therefore, the lack of attraction to MeJA-treated oak trees found in a previous experiment [36] can be explained by the inability of experienced wild birds to associate the volatile blend of MeJA-treated oak trees with any blend of HIPVs Pyrenean oak trees infested with herbivorous prey.

## 5. Conclusions

Overall emission of VOCs produced by MeJA-treated and herbivore-induced trees did not significantly differ from each other. However, MeJA treatment seems to cause a more significant reaction of experimental trees related to the release of several specific compounds compared to herbivore-induced trees. Yet, blends from these two treatments differed from VOCs of Control trees. Whether these slight differences in the emission of volatiles between trees influenced insectivorous bird attraction to MeJA-treated trees remains to be explored. Furthermore, research consensus about the dosage of MeJA, minimal and suitable number of treated leaves per tree, as well as the timing of the volatile measurement after treatment application or infestation should be reached in order to have comparable data among the different studies. The lack of attraction to MeJA-treated and herbivore-induced trees in our study also points out the potential need to perform the experimental studies in periods of low prey availability, or conduct much larger experiments, to increase the relative chances of caterpillars being attacked.

## Figures and Tables

**Figure 1 biology-11-00084-f001:**
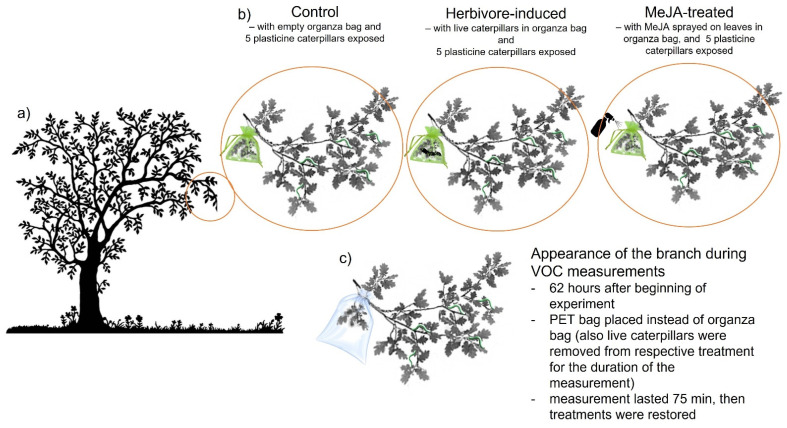
Experiment design showing the approximate location of an experimental branch at each of the focal trees (**a**), three respective treatments as they looked like in the 14 days long predation experiment—Control, herbivore-induced, MeJA-treated (**b**), and appearance of the branches during the measurement of volatile compounds (**c**). The measurement started 62 h after the beginning of the experiment, upon removal of the organza bag and real caterpillars from herbivore-induced treatment. Volatiles were then collected for 75 min. After that, the organza bag and live caterpillars were returned to their respective treatments. For the MeJA application (**b**), the organza bag was always removed, MeJA sprayed on the leaves inside, and the bag returned to the branch.

**Figure 2 biology-11-00084-f002:**
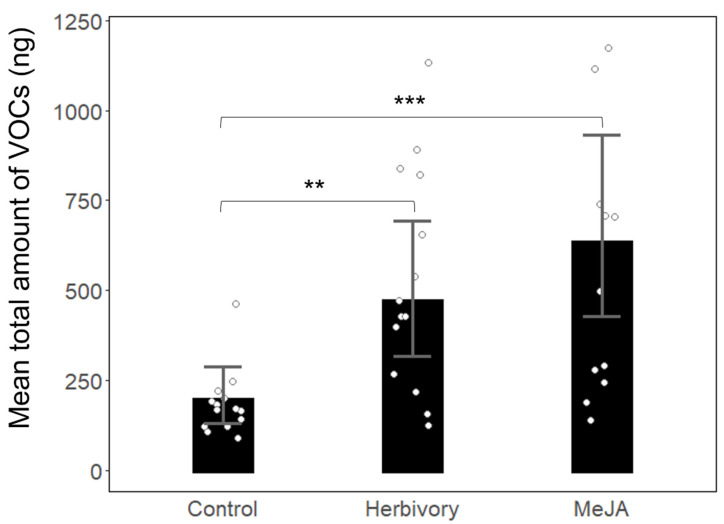
The total amount of volatiles (ng) emitted by Control (*n* = 15), herbivore-induced (*n* = 15), and MeJA-treated (*n* = 15) Pyrenean oaks at a study site in an oak forest of Guadarrama Mountains, Spain. Treatments with significantly different amounts of volatiles are marked by asterisks (*** *p* < 0.001, ** *p* < 0.01); results of chisq test. White circles show VOCs emitted by each individual tree.

**Figure 3 biology-11-00084-f003:**
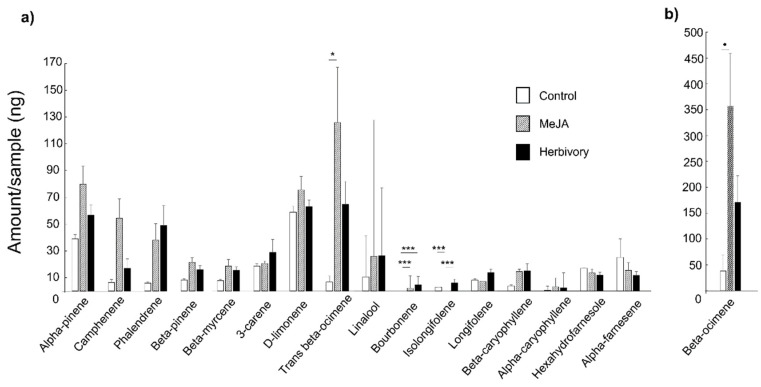
Mean change in the VOCs emission between MeJA-treated, herbivore-induced, and Control trees of Pyrenean oak. The x-axes show the individual compounds, and y-axes show the relative amount (ng; in **a** and **b** with different y-axis scales) of emitted VOCs. MeJA-treated and herbivore-induced trees released significantly higher amounts of Bourbonene (**a**) compared to control trees. MeJA-treated trees emitted more Trans-β-Ocimene and β-Ocimene (**b**) than the control trees. MeJA-treated trees did not produce Isolongifolen in contrast to the control and herbivory-infested trees (**a**). Treatments with significantly different amounts of volatiles are marked by asterisks (*** *p* < 0.001, * *p* < 0.05, ^•^ *p* = 0.05); results of chisq test.

## Data Availability

All data are reported in the manuscript.

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
