# Peer review of "Exogenous Application of Methyl Jasmonate Increases Emissions of Volatile Organic Compounds in Pyrenean Oak Trees, Quercus pyrenaica"

_biology, 2022, doi:10.3390/biology11010084_

Round 1

Reviewer 1 Report

The paper presents some original and experimental work to assess the role of herbivore-induced volatiles on the attraction of birds, and compare volatiles produced after methyl jasmomate application that mimic herbivory damages. Authors took lots of precaution in designing their treatments, controlling for bags, presence of larvae or not on the induction of volatiles. Despite all these efforts, authors did not find any significant effect of odours on bird attraction. Authors provide a lengthy discussion on all possibilities why they did not observe any effect on bird attraction although they did find significant differences in volatiles among the 3 treatments. Overall, the experiment was well designed and authors provided lots of arguments, backed up with appropriate references, why they did not observed any behavioural effect compared to other studies.

There are minor corrections to address, such as:

  • Make sure you use across your text the same terminology, such as ‘herbivore-induced” and not herbivore-infected (this is for pathogen usually), nor capitalised.
  • In the result section, there should not be any interpretation
  • L 373, remove date after referenced name.
  • L 450, replace “whenever” by “whether”
  • 554: please update citation

See other comments in pdf.

I am wondering whether the choice of larval stage (5th instar) may have not been influential, since you would expect in early spring, earlier instars, which may be associated with different volatile induction (apparently first instars of this moth species eat the new leaves, fresh buds, which may be associated with other volatile compounds)? To check, comment.

Also, as discussed by the authors, the treatment with 5mM of MeJa may not be strong enough, or not applied on enough leaves/branches to make the signal stronger.

For the volatile collection, I would have used an active push-pull method instead of passive adsorption, and more than 15 minutes. Any comments?

Did authors collected any visual observation of bird presence in the forest and foraging activity to discuss the absence of attraction to their treated branches? Maybe next time, authors could use some video-cameras to add some behavioural observations.

I found interesting that authors found that herbivore-induced branch produced more isolongifolen, which apparently can be a strong insect repellent ((-)-Isolongifolene = 98.0 GC sum of enantiomers 1135-66-6 (sigmaaldrich.com)), can you discuss this more.

Author Response

Response to comments of Referee 1

The paper presents some original and experimental work to assess the role of herbivore-induced volatiles on the attraction of birds, and compare volatiles produced after methyl jasmomate application that mimic herbivory damages. Authors took lots of precaution in designing their treatments, controlling for bags, presence of larvae or not on the induction of volatiles. Despite all these efforts, authors did not find any significant effect of odours on bird attraction. Authors provide a lengthy discussion on all possibilities why they did not observe any effect on bird attraction although they did find significant differences in volatiles among the 3 treatments. Overall, the experiment was well designed and authors provided lots of arguments, backed up with appropriate references, why they did not observed any behavioural effect compared to other studies.

Response: Thank you very much for your comments.

There are minor corrections to address, such as:

  • Make sure you use across your text the same terminology, such as ‘herbivore-induced” and not herbivore-infected (this is for pathogen usually), nor capitalised.

Response: Thanks, we have used herbivore-induced across the text.

  • In the result section, there should not be any interpretation

Response: We have removed the first part of the first sentence, where we interpret the results.

  • L 373, remove date after referenced name.

Response: Thanks, we have removed it.

  • L 450, replace “whenever” by “whether”

Response: Thank you, we have replaced it.

  • 554: please update citation

Response: Thanks, we have updated the reference.

See other comments in pdf.

I am wondering whether the choice of larval stage (5th instar) may have not been influential, since you would expect in early spring, earlier instars, which may be associated with different volatile induction (apparently first instars of this moth species eat the new leaves, fresh buds, which may be associated with other volatile compounds)? To check, comment.

Response: we captured caterpillars in the same forest where the experiment was performed, therefore, we do not used caterpillars of a different stage that the stages found in the study area in the time when we performed the experiment. Therefore, we do not expect that the use of this larval stage may have affected our results. Of course, as we mentioned in the discussion, the volatiles released by plants may be affected on many factors (Erb & Reymond 2019).

Also, as discussed by the authors, the treatment with 5mM of MeJa may not be strong enough, or not applied on enough leaves/branches to make the signal stronger.

Response: Yes, we agree with you, as we mentioned in the discussion (lines 336-337)

For the volatile collection, I would have used an active push-pull method instead of passive adsorption, and more than 15 minutes. Any comments?

Response: we know that there are different methods for volatile measurement. However, as we measured volatiles in the field, we decided to use our protocol as it was easy to capture volatiles. Furthermore, our methodology has been previously used and demonstrated its efficacy in order to measure plant volatiles (Mrazova & Sam 2017)

Did authors collected any visual observation of bird presence in the forest and foraging activity to discuss the absence of attraction to their treated branches? Maybe next time, authors could use some video-cameras to add some behavioural observations.

Response: We think that birds did not approach the experimental branches, as if they would have approached, they would have probably attacked any artificial caterpillar. So, probable, even if we would have used videocameras, we would probably not obtain any data.

I found interesting that authors found that herbivore-induced branch produced more isolongifolen, which apparently can be a strong insect repellent ((-)-Isolongifolene = 98.0 GC sum of enantiomers 1135-66-6 (sigmaaldrich.com)), can you discuss this more.

Response: yes, isolongifolen can be related to (-)-Isolongifolene, known as an insect repellent. However, the discussion is too long, as pointed out by referee 2, and therefore, we need to reduce it, so we do not have space to discuss more the potential relation between isolongifolen with (-)-Isolongifolene.

Reviewer 2 Report

Review ID Biology-1515813

Exogenous application of methyl jasmonate increases emissions of volatile organic compounds in Pyrenean oak trees, Quercus pyrenaica.

               It is obvious that there is a growing interest in environmentally friendly methods of plant protection in European Union. There is no way for the EU to invest funds in typical chemical plant protection. Large number of pesticides that are ineffective and have negative impact on the environment have been reduced. In crop production, there is a lack of effective pesticides to reduce the population of agraphages. The question is how to protect crops where genetically modified organisms have not been accepted by the EU end user.

               In present time, environmentally friendly methods of plant protection are of importance where natural defense mechanism of plants based on volatile organic compounds may play significant role.

               This is quite well organized manuscript. I found this “ms” interesting and innovative. However, a few questions must be explained more precisely.

Critical review:

  1. Methyl jasmonate is clearly the elicitor of the plant's defense mechanism. But why have other derivatives not been tested for comparison? It is known that even Z-jasmone sometimes induces a stronger response in plants. Please explain.
  2. Lines 35-38.

Volatiles released by MeJa-treated and herbivore infested plants differ or not where several specific compounds after Me-Ja application were observed?

  1. Lines 38-40.

I am not entirely convinced of this statement. Please explain.

  1. Where is the information about communication between plants? What about reactions to typical green leaf volatiles? See papers below.
  2. Lines 66-86.

This entire paragraph seems to be inserted a bit inappropriately. It creates the impression of a place where literature is cited to obtain the effect of 88 items.

  1. Line 81.

(yet [27,28])? Correct.

  1. I do not think it is appropriate to describe the results of the research on one and a half pages. After all, this is the most important part of the work.
  2. The discussion is three pages long. It seems to me that the proportions of the article were not respected.

Some other papers to add:

Volatile organic compounds released by maize following herbivory or insect extract application and communication between plants

Journal of Applied Entomology 141(8), 630-643 (2017)

DOI: 10.1111/JEN.12367

Orientation of European corn borer first instar larvae to synthetic green leaf volatiles

Journal of Applied Entomology 137(3), 234-240 (2013)

DOI: 10.1111/J.1439-0418.2012.01719.X

Author Response

Response of comments to Referee 2

Review ID Biology-1515813

 Exogenous application of methyl jasmonate increases emissions of volatile organic compounds in Pyrenean oak trees, Quercus pyrenaica.

                It is obvious that there is a growing interest in environmentally friendly methods of plant protection in European Union. There is no way for the EU to invest funds in typical chemical plant protection. Large number of pesticides that are ineffective and have negative impact on the environment have been reduced. In crop production, there is a lack of effective pesticides to reduce the population of agraphages. The question is how to protect crops where genetically modified organisms have not been accepted by the EU end user.

               In present time, environmentally friendly methods of plant protection are of importance where natural defense mechanism of plants based on volatile organic compounds may play significant role.

                This is quite well organized manuscript. I found this “ms” interesting and innovative. However, a few questions must be explained more precisely.

Critical review:

  1. Methyl jasmonate is clearly the elicitor of the plant's defense mechanism. But why have other derivatives not been tested for comparison? It is known that even Z-jasmone sometimes induces a stronger response in plants. Please explain.

Response: We focussed the study in comparing the effects of MeJA treatment and herbivore-induced treatment in the attraction of birds and in the volatiles emission of trees because previous studies examining avian attraction to volatiles emitted by herbivore-infested trees and trees treated with phytohormones had used MeJA, therefore, we wanted to obtain comparable results in our system.

  1. Lines 35-38.

Volatiles released by MeJa-treated and herbivore infested plants differ or not where several specific compounds after Me-Ja application were observed?

Response: The overall emission of volatiles produced by MeJA-treated and herbivore-induced trees did not differ, and were higher than emissions of Control trees, although MeJA treatment showed more significant reaction and released several compounds in contrast to herbivore-induced trees: MeJA-treated trees emitted more Trans-β-Ocimene and β-Ocimene than control trees, and MeJA-treated trees did not produce Isolongifolen in contrast with control and herbivory-infested trees (please, see figure 3). We have added in the abstract that these differences were significant.

Lines 38-40.

I am not entirely convinced of this statement. Please explain.

Response: As we have explained across the text, in a previous study we did not find that MeJA treated trees attracted insectivorous birds. Therefore, as there are some differences in the emission of volatiles between MeJA treated trees and herbivore-induced trees, the lack of attraction to MeJA treated trees can be due to such differences, as we explain in detail in the discussion section.

  1. Where is the information about communication between plants? What about reactions to typical green leaf volatiles? See papers below.

Response: In relation to the information about communication between plants, we do not think it is appropriate to mention it as it is quite far from the aim of the study, and results of a study with another Quercus species, Quercus robur, showed that the control branches close to the MeJA-treated branches did not produce more VOCs, i.e. this plant species exhibits a highly localized response (Volf et al. 2021).

  1. Lines 66-86.

This entire paragraph seems to be inserted a bit inappropriately. It creates the impression of a place where literature is cited to obtain the effect of 88 items.

Response: The aim of our study was to study insectivorous bird attraction to Pyrenean oak trees treated with MeJA and to herbivore-induced trees and to find the mechanisms under-lying avian attraction. Therefore, we do not think that explaining in the introduction was it is known about bird attraction to infected trees is inappropriate, we think it is needed as the introduction may reflect the state of the art.

  1. Line 81.

(yet [27,28])? Correct.

  1. I do not think it is appropriate to describe the results of the research on one and a half pages. After all, this is the most important part of the work.

Response: We have explained the results, we do not need more space, as we have explained in detailed the differences found between the treatments in the text and in figure 3.

  1. The discussion is three pages long. It seems to me that the proportions of the article were not respected.

Response: Thanks, we have tried to reduce the length of discussion.

Some other papers to add:

Volatile organic compounds released by maize following herbivory or insect extract application and communication between plants

Journal of Applied Entomology 141(8), 630-643 (2017)

DOI: 10.1111/JEN.12367

Orientation of European corn borer first instar larvae to synthetic green leaf volatiles

Journal of Applied Entomology 137(3), 234-240 (2013)

DOI: 10.1111/J.1439-0418.2012.01719.X

Round 2

Reviewer 2 Report

Literature cited has been not corrected. 88 items still remain. Suggested references not included.

Author Response

Thank you for your comments. We have checked the grammar and corrected some mistakes. We have not included the two suggested references because we think that, although the suggested references report the results of two interesting studies, the aims of such studies are far from the topic of our study, as they deal with the study of communication between plants (Volatile organic compounds released by maize following herbivory or insect extract application and communication between plants. Journal of Applied Entomology 141(8), 630-643 (2017)) and attraction of herbivorous to VOCs (Orientation of European corn borer first instar larvae to synthetic green leaf volatiles. Journal of Applied Entomology 137(3), 234-240 (2013)). As you already pointed out, the manuscript have 88 references that are needed to show the state of the art and to support the discussion of our results, so we do not think that we may decrease the reference list, but also we do not think we have to increase it adding references that are not so related to the study.